# StyleTalker: Stylized Talking Head Avatar from Monocular Video

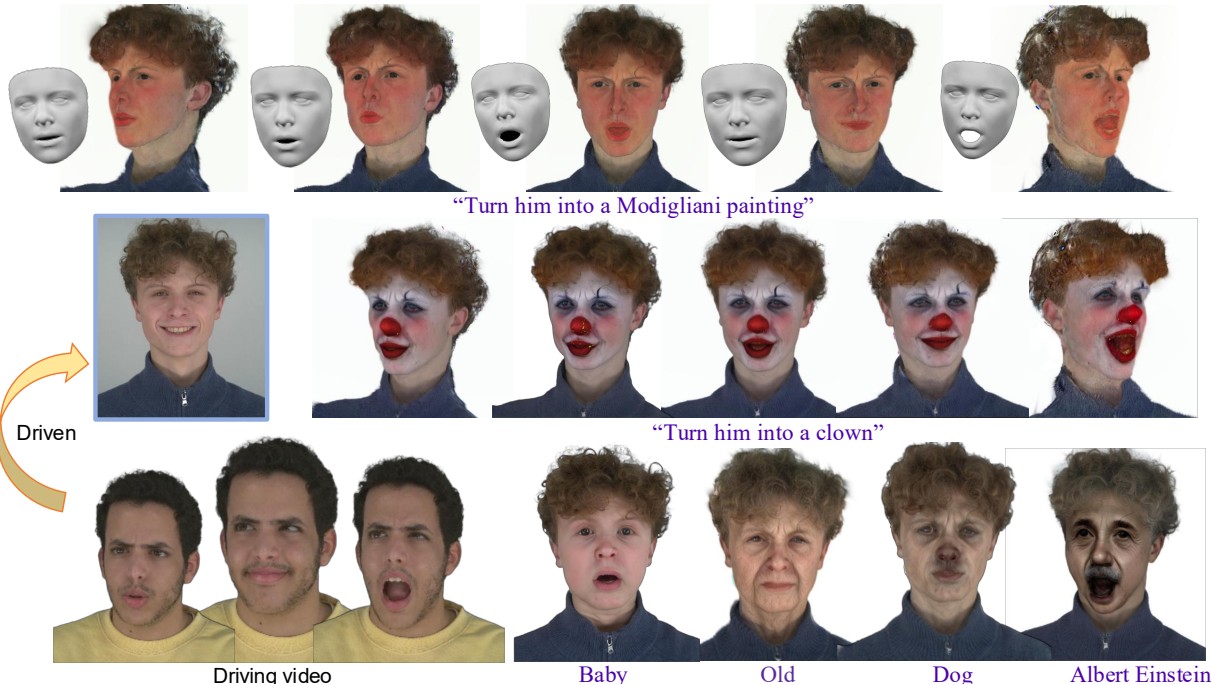

Figure 1: Given a monocular video sequence, StyleTalker can reconstruct a text-controlled 3D stylized digital head which can be animated to different expressions. It not only achieves high-quality editing results but also effectively preserves the original identity.

## Abstract

We introduce **StyleTalker**, a text-guided framework for editing and animating dynamic 3D head avatars from a monocular video. Current 3D scene editing techniques face two main challenges when applied in this task: 1) They typically require multi-view videos for accurate geometry reconstruction. Additionally, they are not suited for dynamic scenarios, making them ineffective for editing talking head avatars from a single-view video. 2) They struggle with fine-grained local edits, largely due to biases inherited from pre-trained 2D image diffusion models and limitations in detecting detailed facial landmarks. To overcome these challenges, we propose StyleTalker with two key innovations: **1)** A **mesh-enhanced 3D Gaussian reconstruction** approach that combines 3D head priors with multi-view video diffusion, improving the accuracy and flexibility of the reconstruction process. **2)** A **landmark-driven talking head editing** method that uses 3D facial landmarks to guide the editing process. By adjusting the strength of the edits based on the distance to these landmarks, our method ensures that the avatar's original identity is preserved while achieving the desired editing. Our extensive experiments demonstrate that StyleTalker outperforms

current state-of-the-art methods, delivering high-quality edits and enabling the animation of avatars with diverse facial expressions, all based on a single-source video.

# 1 Introduction

The modeling and animation of 3D head avatars is a crucial task with significant implications for fields such as digital telepresence, game character design, and augmented/virtual reality (AR/VR). Traditionally, creating realistic 3D head avatars required considerable time and expertise in art and engineering. Recent advancements in deep learning have enabled high-fidelity 3D head reconstruction and animation. However, these methods typically rely on dense multi-view video inputs, limiting their applicability in lightweight or real-time applications.

Recent developments in neural rendering techniques, particularly neural radiance fields (NeRF) Mildenhall et al. (2020) and 3D Gaussian Splatting (3DGS) Kerbl et al. (2023), have significantly propelled the state of the art in reconstructing animatable 3D avatars from monocular video inputs Yu et al. (2023); Xiang et al. (2024); Jiang et al. (2023). However, as the quality of these avatars approaches lifelike detail, concerns regarding privacy and identity security, particularly in virtual environments like the metaverse, become increasingly important. This underscores the urgent need for editable talking head avatars that respect both the identity and desired changes in expression and appearance. In this work, we take the initiative and showcase the potential of addressing this challenge by leveraging the 3D generative methods in Fig. 1.

Recently, the availability of large datasets Xia et al. (2021); Rostamzadeh et al. (2018) containing extensive text-image pairs have driven significant advances in vision-language models (e.g., CLIP Radford et al. (2021)) and diffusion models (e.g., Stable Diffusion Stability.AI (2022); Saharia et al. (2022); Rombach et al. (2022)). These advancements have led to the emergence of text-guided 3D scene editing methods Brooks et al. (2023); Park et al. (2024); Haque et al. (2023); Dong & Wang (2023); Chen et al. (2024a); Wu et al. (2024). A notable approach is Instruct-NeRF2NeRF (IN2N) Haque et al. (2023), which uses an image-conditioned diffusion model Brooks et al. (2023) to iteratively edit images rendered from NeRF. Subsequent works have built on this idea of editing 3D scenes via 2D renderings and improve IN2N from various angles, including improving visual quality Dong & Wang (2023), incorporating 3D Gaussian Splitting (3DGS) for better efficiency Wu et al. (2024), and enabling more controllable and localized editing Chen et al. (2024a).

Considering the flexibility and strong performance of IN2N and its follow-up works, it might seem that these 3D scene editing methods are adequate for editing talking head avatars while maintaining the original identity. However, existing methods have two key limitations (see Fig. 4): *1) Dependence on multi-view or static video inputs.* Existing text-guided 3D scene editing methods rely on multi-view images for the COLMAP Schönberger & Frahm (2016) reconstruction process and overlook dynamic scenarios, making them unsuitable for editing dynamic talking head avatars from a monocular video. *2) Limitations in fine-grained local editing.* Although GaussianEditor Chen et al. (2024a) enables local editing by targeting specific areas using the large language model (*i.e.*, Lang-SAM Kirillov et al. (2023)), we find that this approach is still prone to identity loss. This problem arises from two main causes: (i) Lang-SAM struggles to accurately locate detailed facial landmarks, which are crucial for preserving identity, and (ii) the image diffusion model used can produce varying levels of editing strength across iterations, resulting in inconsistent gradient back-propagation.

To address the challenges, we introduce a novel 3D scene editing method, StyleTalker, designed to edit talking head avatars from monocular videos. Our approach consists of two key components: **1) Mesh-enhanced 3D Gaussian reconstruction.** To achieve high-quality 3D reconstructions from a single-view video, we propose a two-stage training strategy that integrates both 3D head priors and multi-view video diffusion models into the reconstruction pipeline. **2) Landmark-driven talking head editing.** To overcome the difficulties of localized editing while preserving the original identity, we introduce landmark-driven editing. Specifically, we leverage 3D landmarks as a guide to control the editing strength across different facial regions. By adjusting the editing strength based on the distance to landmarks, we successfully ensure the original identity is preserved while following the editing instructions. Our method also allows the edited avatars to be animated with different expressions, driven by a source video.

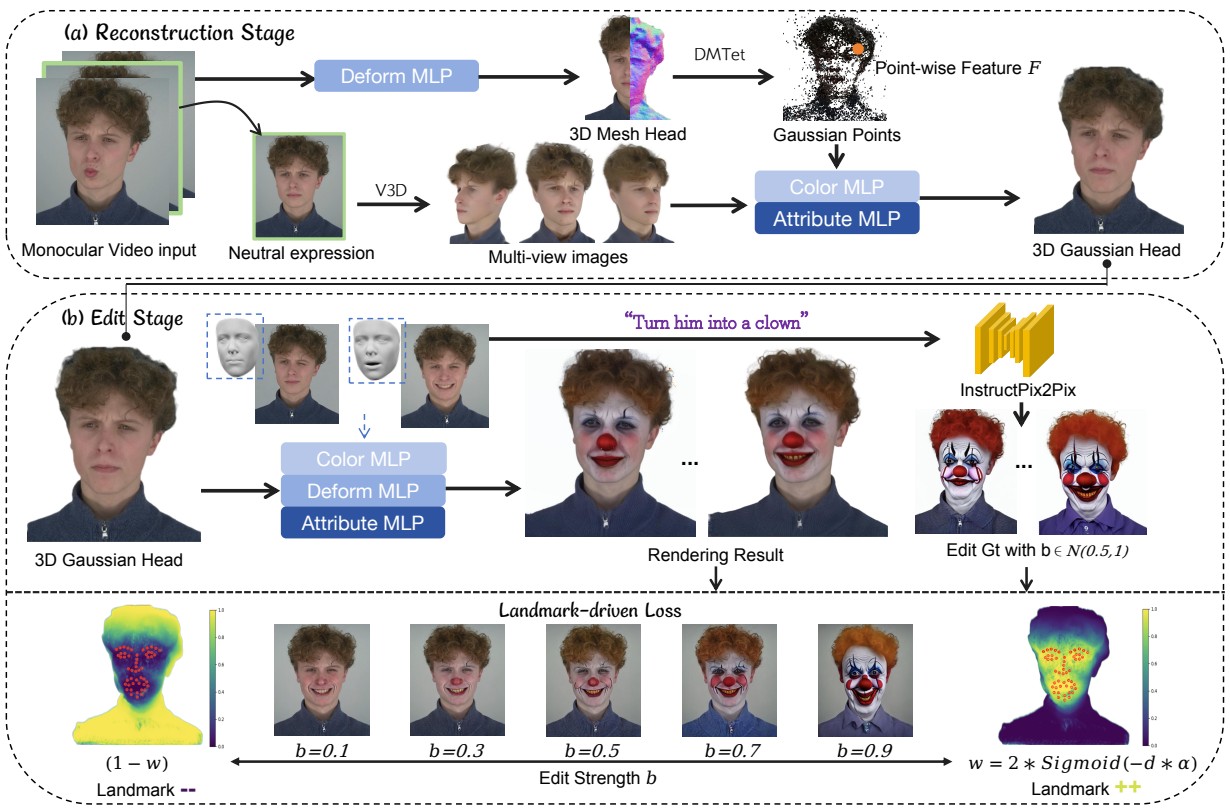

Figure 2: **Overview of StyleTalker.** Our method takes as input a monocular video and text prompts to edit dynamic 3D scenes represented by 3D Gaussian Splatting. **(1)** We first integrate both 3D head prior and multi-view video diffusion model (V3D) to reconstruct the detailed 3D Gaussian head. **(2)** We propose a weighted loss related to edit strength $b$ and the 3D facial landmarks, effectively achieve local editing while maintaining the original character identity.

We conduct a comprehensive evaluation of StyleTalker by applying it to edit a variety of talking head avatars using different text prompts (see Fig. 1). The experimental results demonstrate that StyleTalker achieves high-quality edited 3D geometry and texture from monocular videos while preserving the identity of the avatars through local edits. This performance significantly surpasses that of existing methods. Furthermore, our edited avatars can be animated with diverse expressions based on a driven video.

## 2    Related Work

**Text-guided image generation and editing.**    Recent advances in vision-language models, such as CLIP Radford et al. (2021), and diffusion models Ho et al. (2020); Dhariwal & Nichol (2020); Rombach et al. (2022); Song et al. (2021), have enabled the generation of detailed 2D content from text prompts, driven by large-scale text-image datasets Schuhmann et al. (2022; 2021). Models such as DALL · E 2 Ramesh et al. (2022), Imagen Saharia et al. (2022), and Stable Diffusion Stability.AI (2022) create realistic images from textual descriptions. Building on this, research has focused on refining the control of results  Zhang & Agrawala (2023); Zhao et al. (2023); Mou et al. (2023), extending diffusion models to video generation Singer et al. (2023); Blattmann et al. (2023), and enabling diverse editing tasks for images and videos Hertz et al. (2022); Kawar et al. (2023); Wu et al. (2023); Brooks et al. (2023); Valevski et al. (2022); Esser et al. (2023); Hertz et al. (2023). Efforts to improve content generation for personalized subjects include DreamBooth Ruiz et al. (2023) and textual inversion Gal et al. (2023). Despite the significant progress made in text-to-image and text-to-video generation, achieving the desired outcome from text prompts remains a challenge, particularly

for editing tasks. One of the key difficulties is the inherent unpredictability of these models. In the case of head avatars, this randomness often leads to the loss of the original facial expressions, making it difficult to maintain consistency with the source avatar while editing.

**Text-guided 3D scene editing.** With the progress in differentiable 3D representations, such as NeRF Mildenhall et al. (2020) and 3DGS (Kerbl et al., 2023), and the progress in diffusion-based text-to-2D generation models Stability.AI (2022); Brooks et al. (2023), text-driven 3D scene editing methods have emerged. These methods aim to modify 3D objects or scenes simply based on a textual description. One of the earliest attempts in this area, Instruct-NeRF2NeRF (IN2N) Haque et al. (2023), proposes leveraging Instruct-Pix2Pix(Brooks et al., 2023) to edit 2D renderings and backpropagates gradients to edit the 3D scene. While IN2N shows potential, it suffers from several issues, including instability, inefficient training, blurry results, and geometry distortion. To address these limitations, subsequent works Po et al. (2024); Wang et al. (2024) have aimed to enhance performance from various angles: Instruct-Gaussian2Gaussian Cyrus & Ayyan (2023) replaces NeRF with 3DGS and introduces more effective dataset updating strategies to boost training efficiency. Vica-NeRF Dong & Wang (2023) proposes selecting reference images from the input dataset, edit them with Instruct-Pix2Pix, and blend the results across the rest of the dataset to reduce inconsistencies. However, this blending technique often results in blurry edits, particularly with human subjects, and does not fully address the consistency issue. Other methods, such as DreamEditor Zhuang et al. (2023), utilize personalized models like DreamBooth Ruiz et al. (2023) for localized editing, while TIP-Editor Zhuang et al. (2024) introduces a 3D bounding box as a condition to enhance control over specific areas in the scene. Despite these improvements, such techniques still struggle with modifying internal geometry and textures in a way that maintains consistency and realism. Finally, approaches such as GaussianEditor Chen et al. (2024a) and GaussCTRL Wu et al. (2024) utilize large language models Kirillov et al. (2023) and depth-conditioned ControlNet Zhang & Agrawala (2023) to enable more precise, text-driven local editing. Although these methods show promise in some contexts, they heavily rely on "multi-view" "static" videos as inputs to reconstruct the 3D geometry via COLMAP Schönberger & Frahm (2016). These limitations make it challenging for them to edit talking head avatars from a monocular video.

**3D head reconstruction.** Reconstructing 3D head avatars from monocular videos is both a popular and challenging research area. Early approaches Cao et al. (2015; 2016); Ichim et al. (2015); Hu et al. (2017); Deng et al. (2019); Nagano et al. (2018) focused on optimizing a morphable mesh to fit the video data. More recent methods Grassal et al. (2022); Khakhulin et al. (2022) use neural networks to model non-rigid deformations over 3D morphable face models (3DMM)Li et al. (2017b); Gerig et al. (2018b), allowing for the recovery of more dynamic facial details. However, these methods still lack the flexibility to handle complex topological variations in the geometry. As a result, newer techniques have begun exploring more advanced representations, such as implicit Signed Distance Fields (SDF) Zheng et al. (2022), point clouds Zheng et al. (2023), and Neural Radiance Fields (NeRF) Guo et al. (2021); Liu et al. (2022); Gafni et al. (2021); Athar et al. (2023; 2022); Gao et al. (2022); Xu et al. (2023a); Zielonka et al. (2023); Xu et al. (2023b); Qin et al. (2024), which offer better handling of detailed, complex head structures and dynamic features.

## 3 Methodology

Given a single-view video of a talking head avatar, we begin by preprocessing each frame with the Basel Face Model (BFM) Schönberger & Frahm (2016) to extract the head pose $\theta$ and expression $\beta$ coefficients. As depicted in Fig. 2, our approach proceeds in two main stages: (1) high-quality reconstruction using both mesh and 3D Gaussian Splatting, and (2) editing and animation through a pre-trained diffusion model. In the following sections, we first introduce the foundational concepts underlying our method in Sec. 3.1. We then describe the key components of StyleTalker, including (1) mesh-enhanced 3D Gaussian reconstruction via multi-view video diffusion model Chen et al. (2024b) in Sec. 3.2, and (2) landmark-driven talking head editing for identity-preserving head editing in Sec. 3.3.

### 3.1 Preliminaries

**3D Gaussian Splatting.** 3D Gaussian Splatting (3DGS) Kerbl et al. (2023) directly optimizes the 3D positions $\mathbf{x}$ and attributes of 3D Gaussians. Each 3D Gaussian, denoted as $G(\mathbf{x})$, is defined by a 3D covariance matrix $\Sigma$ and a mean (center) at $\mu$:

$$G(\mathbf{x}) = e^{-\frac{1}{2}(\mathbf{x}-\mu)^T \Sigma^{-1}(\mathbf{x}-\mu)}. \tag{1}$$

To improve rendering efficiency, 3DGS utilizes a tile-based rasterizer Lassner & Zollhofer (2021). The screen is partitioned into tiles, typically sized $16 \times 16$ pixels, with each Gaussian instantiated according to the number of overlapping tiles. A key is assigned to each Gaussian, which stores information about its depth in view space and its corresponding tile ID. These Gaussians are then sorted by depth to handle occlusions and overlapping geometries during the rasterization process correctly. 3DGS employs a point-based $\alpha$-blend rendering technique to compute the RGB color $\mathbf{C}$ for the rendering, by sampling points along a ray at intervals $\delta_i$:

$$\mathbf{C}_{\text{color}} = \sum_{i \in N} \mathbf{c}_i \sigma_i \prod_{j=1}^{i-1} (1 - \sigma_j), \quad \sigma_i = \alpha_i e^{-\frac{1}{2}(\mathbf{x})^{\mathrm{T}} \mathbf{\Sigma}^{-1}(\mathbf{x})}, \tag{2}$$

where $\mathbf{c}_i$ is the color of each point along the ray and $\sigma_i$ represents the opacity.

**3D scene editing.** Building upon techniques like Instruct-Pix2Pix Brooks et al. (2023) and NeRF Mildenhall et al. (2020)/3DGS Kerbl et al. (2023), current methods for 3D scene editing typically follow a two-step process to achieve their desired outcomes:

*1) Image editing.* Given a rendered image from a specific camera viewpoint for each iteration, the network will first introduce Gaussian noise to the image. This noisy image, along with the text embedding $y$ and the original training image, is used as input for Instruct-Pix2Pix, which generates an edited version reflecting the desired changes. These modifications are then back-propagated to the 3DGS scene to update the 3D model accordingly.

*2) Dataset update.* In addition to back-propagating the editing direction, the network will also periodically update the entire dataset, typically every $2,500$ training iterations. This update involves replacing the original images with their edited versions, ensuring progressively stronger and more accurate 3D edits over time.

### 3.2 Mesh-enhanced 3D Gaussian reconstruction

Existing 3D scene editing methods typically rely on multi-view images as input, which enables the use of COLMAP Schönberger & Frahm (2016) to extract camera calibrations and reconstruct the original 3D scene. Additionally, current techniques Haque et al. (2023); Chen et al. (2024a); Wu et al. (2024) often assume a static scene. However, these assumptions can restrict the method's applicability in real-world scenarios, particularly in cases such as talking head videos, where multi-view images are rarely available. Given the rapid advancements in generating multi-view videos from single-view inputs, it might be assumed that applying these models to create multi-view videos could effectively solve the problem. However, we observe that current techniques Xie et al. (2024) still struggle with head-related tasks, making it impractical to simultaneously train both deformable geometry and texture at the same stage. To overcome these challenges, we propose a two-stage training strategy that integrates both 3D head priors and multi-view video diffusion models into the reconstruction pipeline.

**Mesh-based deformation field.** In Sec. 3.1, we describe the strengths of 3DGS. Nevertheless, we note that directly training the deformation field using 3DGS would necessitate a highly detailed initialization. To address this, we follow Gaussian-Head-Avatar Xu et al. (2024) and use DMTet Shen et al. (2021) to first optimize the 3D deformable geometry. Specifically, we initialize DMTet using a unit sphere, and for each point $\mathbf{x}$, we predict both its color ($\mathbf{c}$) and position ($\mathbf{pos}$) as follows:

$$\mathbf{c_x} = \hat{\text{MLP}}_{\text{color}}(f, \beta), \quad \mathbf{pos_x} = \text{MLP}_{\text{deform}}(f, \gamma(\mathbf{x})), \tag{3}$$

where $\hat{\mathtt{ML}}\mathtt{P}_{\text{color}}$ and $\mathtt{MLP}_{\text{deform}}$ are MLPs to respectively predict the color and position value. $f$ denotes a learnable feature that is randomly initialized, which serves as the connection between color and geometry to improve the reconstruction quality. $\beta$ represents the expression coefficients predicted via BFM Gerig et al. (2018a), and $\gamma(\cdot)$ is the pose embedding function.

To learn the deformation field $\mathtt{MLP}_{\text{deform}}$, we supervise the training via:

$$\mathcal{L} = \lambda_{\text{rgb}} \cdot \mathcal{L}_{\text{RGB}} + \lambda_{\text{def}} \cdot \mathcal{L}_{\text{def}} + \lambda_{\text{sil}} \cdot \mathcal{L}_{\text{sil}} + \lambda_{\text{lmk}} \cdot \mathcal{L}_{\text{lmk}}, \tag{4}$$

$$\mathcal{L}_{\text{RGB}} = ||\mathbf{C}_{\text{render}} - \mathbf{C}_{\text{gt}}||_1, \quad \mathcal{L}_{\text{def}} = ||\mathbf{P} - \mathbf{P}_{\text{gt}}||_2, \quad \mathcal{L}_{\text{sil}} = \mathcal{IOU}(\mathbf{M}, \mathbf{M}_{\text{gt}}), \tag{5}$$

where $\mathbf{C}_{\text{gt}}$ and $\mathbf{M}_{\text{gt}}$ respectively denote the ground-truth RGB image and mask from the input monocular video, $\mathbf{P}$ represents the predicted expression conditioned landmarks, $\mathcal{IOU}$ denotes Intersection over Union metrics, and $\mathcal{L}_{\text{lmk}}$ is a regularization term that ensures the SDF values at the 3D landmarks are close to zero, thereby positioning the landmarks on the surface of the mesh.

**Diffusion-guided color pre-training.** Although the above optimization learns color information, it struggles with producing view-consistent textures. This limitation arises because the training images used for DMTet only include frontal views, leading to poor performance when the model is rotated to other angles. To address this, we propose leveraging a pre-trained video diffusion model (*i.e.*, V3D Chen et al. (2024b)) to optimize a 3DGS scene. Specifically, we begin by extracting the mesh from DMTet, which provides the vertex positions $\mathbf{x}$, per-vertex feature vectors $f$, and the mesh faces. These components serve as the initial geometry for 3DGS optimization. Next, we randomly select a frame from the input video that depicts a neutral expression. We then use the pre-trained V3D model to generate multi-view images, which are used to train the model:

$$\mathbf{c}_{\mathbf{x}} = \mathtt{MLP}_{\text{color}}(f, \beta), \quad \mathbf{pos}_{\mathbf{x}} = \mathtt{MLP}_{\text{deform}}(f, \gamma(\mathbf{x})), \quad \text{Attri}_{\mathbf{x}} = \mathtt{MLP}_{\text{Attri}}(f, \beta), \tag{6}$$

where $\text{Attri}_{\mathbf{x}}$ denotes the attributes of 3D Gaussian point $\mathbf{x}$.

Importantly, we train the color MLP $\mathtt{MLP}_{\text{color}}$ and attribute MLP $\mathtt{MLP}_{\text{attri}}$ from scratch to ensure the color consistency of the textures, while we retain the pre-trained weights for the deformation MLP $\mathtt{MLP}_{\text{deform}}$. This design helps mitigate the risk of geometry distortion, as the generated multi-view images may still exhibit some inconsistency, particularly when transitioning between different viewpoints (see Fig. 8). Note that the model is supervised via generated multi-view images with the same loss functions as in Eq. 4.

### 3.3 Landmark-driven talking head editing

Once the reconstruction is obtained, editing it locally to achieve the desired changes while preserving the original identity presents an additional challenge. GaussianEditor Chen et al. (2024a) has tackled this by using a large language model (Lang-SAM Kirillov et al. (2023)). However, when applied to head avatars, Lang-SAM tends to treat the entire head as a single segment, failing to distinguish smaller, critical features such as facial landmarks that are key to a person's identity. This limitation makes it difficult for GaussianEditor to maintain the original identity during edits.

To tackle this challenge, one might consider using ControlNet-based Instruct-Pix2Pix Brooks et al. (2023) as in HeadSculpt, which might appear to be a promising solution. However, this approach is prone to inheriting biases from the pre-trained diffusion model. These biases arise because, during each iteration, the employed diffusion model (*e.g.*, Instruct-Pix2Pix) will randomly apply an editing strength, denoted as $b$, with $b$ sampled from a Gaussian distribution (*i.e.*, $b \sim \mathcal{N}(0.5, 1)$). Typically, when $b$ exceeds 0.7, the edited image loses most of the original identity. While setting $b$ to a constant value may seem like a potential fix, it leads to two key issues: **1)** insufficient editing, and **2)** unwanted modifications to areas such as clothing or other untargeted regions (see Fig. 6).

To this end, we propose a landmark-driven approach to talking head editing, which controls the strength of edits applied to different regions by adjusting the editing strength parameter $b$ locally according to the 3D point's position $\mathbf{x}$. Specifically, given an editing instruction $y$ and an input image $\mathbf{C}_{\text{gt}}$, we first pass them

Table 1: **Quantitative comparisons.** CLIP numbers are calculated via CLIP-L/14. "GE": GaussianEditor; "GC": GaussCtrl.

|  | IN2N (GS) (Cyrus & Ayyan, 2023) | GE (Chen et al., 2024a) | GC (Wu et al., 2024) | Ours |
|---|---|---|---|---|
| User Study | 10.0% | 13.3% | 3.3% | **73.3%** |
| CLIP-DS ↑ | 0.60 | 0.63 | 0.59 | **0.68** |
| CLIP-S ↑ | 0.27 | 0.29 | 0.30 | **0.32** |

Table 2: **Quantitative comparisons under the same setting with multi-view videos.** CLIP numbers are calculated via CLIP-L/14. "GE": GaussianEditor; "GC": GaussCtrl.

|  | IN2N (GS) (Cyrus & Ayyan, 2023) | GE (Chen et al., 2024a) | GC (Wu et al., 2024) | Ours |
|---|---|---|---|---|
| User Study | 12.6% | 17.1% | 9.6% | **60.7%** |
| CLIP-DS ↑ | 0.60 | 0.56 | 0.64 | **0.68** |
| CLIP-S ↑ | 0.23 | 0.31 | 0.28 | **0.32** |

through Instruct-Pix2Pix to generate an edited image $\mathbf{C}_{\text{edit}}$. Following Wu et al. (2024); Chen et al. (2024a), we then adopt the $\ell_1$ loss:

$$\mathcal{L}_{\text{edit}} = ||\mathbf{C}_{\text{edit}} - \mathbf{C}_{\text{render}}||_1, \tag{7}$$

where $\mathbf{C}_{\text{render}}$ is the rendered image.

However, directly back-propagating this loss leads to the issues mentioned earlier. To resolve this, for each 3D Gaussian point $\mathbf{x}$, we calculate its distance $d$ (with $d > 0$) to the nearest 3D landmark and then compute a weight map $w = 2 \cdot \texttt{sigmoid}(-d \cdot \alpha)$, where $\alpha$ is a hyperparameter. This allows our landmark-driven editing method to adjust the loss function as follows:

$$\mathcal{L}_{\text{edit}} := \begin{cases} \mathcal{L}_{\text{edit}} \cdot w, b >= 0.5 \\ \mathcal{L}_{\text{edit}} \cdot (1 - w), b < 0.5. \end{cases} \tag{8}$$

We set $\alpha = 5$ for our experiments.

# 4 Experiments

We now assess the performance of StyleTalker across various scenarios and provide a comparative analysis with state-of-the-art 3D scene editing pipelines.

**Implementation details.** We build our network based on the implementation of Gaussian-Head-Avatar Xu et al. (2024). Specifically, we first preprocess the data by removing the background and extracting 64-dimensional 2D facial landmarks from all images. We then extract the 3D landmarks, expression coefficients $\beta \in \mathbb{R}^{64}$, head pose $\theta \in \mathbb{R}^3$ by passing all frames into the Basel Face Model (BFM) Gerig et al. (2018a). For each input monocular video, we select a frame with a neutral expression using V3D Chen et al. (2024b) to obtain multi-view information from six cameras positioned approximately 120 degrees apart in the front. We collect 10 data sets from NerSemble Kirschstein et al. (2023) to perform the experiments provided in the paper. For each experiment, we train the mesh-based deformation field for 10,000 iterations (around 10 minutes), followed by the diffusion-guided color pre-training for 2,000 iterations (approximately 5 minutes). Afterward, we train the edit stage for 20,000 iterations with a batch size of 1 until full convergence, which takes around 60 minutes. All experiments are run on a single 4090 GPU using the Adam Kingma & Ba (2015) optimizer, with a learning rate of 0.0001. The hyperparameters are set to $\lambda_{\text{rgb}} = 0.1$, $\lambda_{\text{def}} = 1$, $\lambda_{\text{sil}} = 0.1$, $\lambda_{\text{lmk}} = 0.01$.

**Baseline methods.** We compare our editing results with three baselines: GaussianEditor Chen et al. (2024a), IN2N(GS) Haque et al. (2023), and GaussCTRL Wu et al. (2024). We do not include IN2N and

Table 3: **Quantitative comparisons regarding the reconstruction quality.**

| | PointAvatar Zheng et al. (2023) | FlashAvatar Xiang et al. (2024) | INSTA Zielonka et al. (2023) | Gaussian-Head-Avatar Xu et al. (2024) | Ours | Ours (18 views) |
|---|---|---|---|---|---|---|
| LPIPS ↓ | 0.086 | 0.075 | 0.071 | **0.062** | 0.070 | 0.065 |
| SSIM ↓ | 0.833 | 0.877 | 0.879 | 0.884 | 0.884 | **0.888** |
| PSNR ↑ | 24.48 | 27.41 | 26.77 | **27.59** | 27.50 | 27.57 |

Vica-NeRF in the comparison, as they have been shown to perform similarly or worse than the methods mentioned above. Additionally, we cannot compare with TextToon Song et al. (2024) due to the unavailability of their code.

## 4.1 Quantitative evaluations

**User studies.** We conducted user studies to compare our method against three baseline approaches Cyrus & Ayyan (2023); Chen et al. (2024a); Wu et al. (2024). A total of 30 volunteers viewed 15 rotating videos generated by different methods and were asked to select the one they preferred the most. The results, summarized in Tab. 1 and Tab. 2, show that our method received the highest preference overall.

**CLIP-based metrics.** Building on the approach from GaussCtrl Wu et al. (2024), we compute two metrics to assess editing performance: CLIP-Score (CLIP-S Hessel et al. (2021) and CLIP Directional Similarity (CLIP-DS) Brooks et al. (2023); Gal et al. (2022). These metrics are calculated using 15 results, and the numbers are presented in Tab. 1 and Tab. 2. The results demonstrate the superiority of our framework, emphasizing its enhanced editing fidelity and superior preservation of identity when compared to alternative methods

**Reconstruction quality.** We further conduct additional quantitative evaluations of reconstruction quality. Specifically, we use 7 examples from the NeRSermble dataset to calculate the numbers. The results, presented in Tab. 3, demonstrate that (1) our method achieves slightly better performance compared to state-of-the-art reconstruction techniques, and (2) our method performs on par with Gaussian-Head-Avatar that requires multi-view videos as the input. Note that the values are calculated based on the ground-truth monocular video and corresponding renderings from the reconstructed head.

**Analysis of the number of views by V3D Chen et al. (2024b).** In our approach, we employ V3D to generate 6-view multi-view images as priors for high-quality reconstruction, striking a balance between quality and efficiency. To further explore the potential of V3D, we also generate 18 views to assess whether additional views can enhance reconstruction quality. We conducted *a user study* with

Table 4: **Quantitative comparisons regarding the reconstruction quality.**

| | Ours (6 views) | Ours (18 views) |
|---|---|---|
| Overall quality ↑ | 38.9% | **61.1%** |
| Absence of artifacts ↑ | 24.6% | **75.4%** |

27 volunteers. Each participant is required to compared 15 pairs of rotating results and selected the better one based on two criteria: a) overall quality and b) absence of artifacts. The statistical results, shown in Tab. 4, indicate that increasing the number of views from 6 to 18 yields a clear improvement in reconstruction quality. (2) We also show the quantitative evaluations in Tab. 3. By increasing the number of views from 6 to 18 yields, we can observe consistent improvements across different metrics.

## 4.2 Qualitative evaluations

IN2N(GS)  GaussianEditor  GaussCtrl  Ours

"Turn him into a clown"

"Turn him into a dog"

"Turn him into a Van Gogh Style"

Original Look  "Turn him into a Modigliani painting"

Figure 4: **Comparison with existing 3D scene editing methods.** Unlike other methods that struggle or fail to preserve the original identity and facial expressions, our method consistently addresses these issues and delivers superior results.

**Talking head editing under various scenarios.** In Fig. 1, we present a diverse set of 3D talking head avatars edited using our method. The results consistently demonstrate identity-preserving edits with high-quality geometry and texture, viewed from multiple angles.

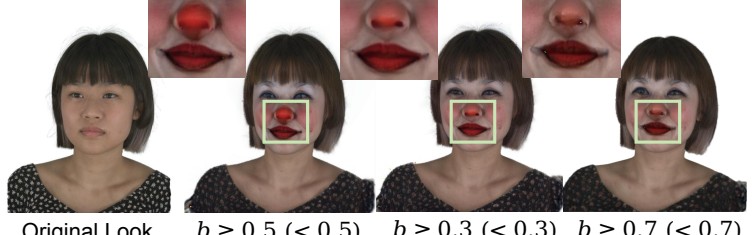

Original Look  $b \geq 0.5 \ (< 0.5)$  $b \geq 0.3 \ (< 0.3)$  $b \geq 0.7 \ (< 0.7)$

Figure 3: **Different edit results of applying different boundaries for the editing strength $b$.**

**Talking head editing with different animation.** Our method pre-trains a deformation field $\texttt{MLP}_{\text{deform}}$ and incorporates the face expression parameters $\beta$ during the reconstruction stage. This enables us to animate the edited head avatars using expressions from any source video. To showcase this capability, we present several animated examples in Fig. 1. The results demonstrate that our method produces high-quality animations that align with the driving video while preserving the original edits.

Table 5: **Quantitative comparisons regarding the ablation studies.**

|  | *w/o* V3D Chen et al. (2024b) | *w/o* mesh-enhanced deformation field (Sec. 3.2) | Ours |
|---|---|---|---|
| LPIPS ↓ | 0.083 | 0.335 | **0.070** |
| SSIM ↓ | 0.870 | 0.786 | **0.884** |
| PSNR ↑ | 26.11 | 21.51 | **27.50** |

Figure 5: **Different edit results by modifying the control weight** $w$.

**Talking head editing with different edit scale.** In Fig. 5, we demonstrate that our method allows for adjustable editing levels by modifying the control weight $w$. Furthermore, the intensity of the editing can be fine-tuned by adjusting the boundary of $b$ in Eq.8. For instance, we set $b >= 0.3(b < 0.3)$, $b >= 0.5(b < 0.5)$, and $b >= 0.7(b < 0.7)$ as the boundary for experiments and present the corresponding results in Fig. 3.

**Comparisons with baseline methods.** We present qualitative comparisons with existing methods in Fig. 4. The following key insights can be drawn from the visual results: **(1)** All baseline methods tend to distort the original identity in their edited outputs. **(2)** These methods often alter the character's clothing or the background, which is undesirable in most cases. **(3)** The baseline methods fail to preserve the original facial expressions, frequently resulting in neutral expressions in the edited images. In contrast, our method consistently delivers superior results by preserving both identity and expression, along with high-quality geometry and texture. Additional comparisons are available in the supplementary material.

### 4.3 Ablation studies

**Effectiveness of mesh-based deformation field.** We begin by conducting ablation studies to evaluate the effect of the mesh-based deformation field, with the results shown in Fig. 7, Fig. 8, and Tab. 5. Specifically, we compare two variants: (**D1**) We initialize the 3D Gaussian Surface (3DGS) from a unit sphere and perform the 3D reconstruction directly using the proposed diffusion-guided color pre-training. (**D2**) We initialize the 3DGS from a well-trained DMTet model but train the MLP$_{deform}$ from scratch during the diffusion-guided color pre-training. The comparing **D1** and **D2** with our final results shows that: (1) Without the proposed mesh-based deformation field, the 3D reconstruction suffers from significant geometric distortions that negatively impact the quality of results; (2) the pre-trained MLP$_{deform}$ plays a crucial role in preserving facial expressions in the edited results.

**Effectiveness of multi-view video diffusion model.** In Fig. 8 and Tab. 5, we present additional ablation studies to highlight the effectiveness of the multi-view video diffusion model. The visual results demonstrate that without V3D, the 3D reconstruction fails to produce high-quality results when observed from side views.

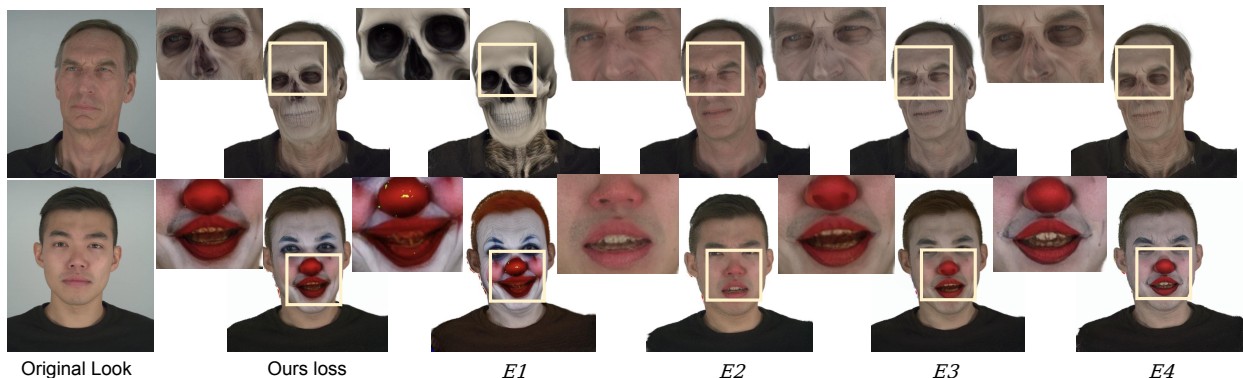

Figure 6: **Different edit results with different loss functions.**

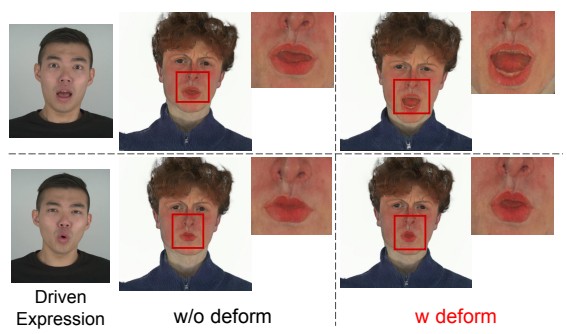

Figure 7: **Effectiveness of mesh-based deformation field.**

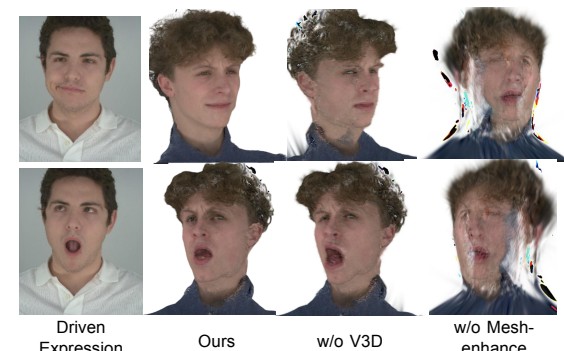

Figure 8: **Analysis of mesh-enhanced reconstruction.**

**Effectiveness of landmark-driven talking head editing.** In Fig. 6, we show edited results using different configurations of loss functions. Among them, **(E1)** applies the original loss function $\mathcal{L}_{\text{edit}}$ as in Eq. 7, with a fixed editing strength of $b = 0.8$; **(E2)** follows the same setup as **(E1)** uses a smaller editing strength with $b = 0.2$; **(E3)** also starts with **(E1)** but sets the editing strength randomly, following a Gaussian distribution; **(E4)** applies a modified loss function $\mathcal{L}_{\text{edit}} := \mathcal{L}_{\text{edit}} \cdot w$. From the results shown in Fig. 6, we make the following observations: **(1)** when the editing strength is fixed at a large value, the original identity of the avatar is lost; **(2)** a small editing strength will instead lead to insufficient editing; **(3)** both **E3** and **E4** will lead to insufficient editing, failing to produce the desired level of modification.

## 5 Conclusion

We introduce StyleTalker, a novel pipeline for editing and animating 3D talking heads from a monocular video. Our approach begins with a two-stage training strategy, incorporating mesh-enhanced 3D Gaussian reconstruction to integrate 3D head priors and multi-view video diffusion models, thus enabling high-quality deformable meshes from a single-view video. We then introduce landmark-driven talking head editing that preserves the original identity while performing fine-grained editing. Extensive evaluations demonstrate that our method delivers impressive results across a wide range of scenarios, outperforming current state-of-the-art methods.

**Limitations.** While demonstrating the potential for editable talking head, we recognize that StyleTalker has certain limitations: **1)** It inherits constraints from InstructPix2Pix Brooks et al. (2023), such as the inability

to perform large-scale spatial edits (e.g., removing a nose). **2)** The editing process is not yet real-time, our method still requires a non-negligible amount of time to generate the edited talking avatar.

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

## Appendix

## A  Further illustration

**More description of landmark loss.**   Here, we provide a detailed explanation of how the landmark loss works during training. Specifically, at each iteration:

- The edit strength is randomly sampled from a Gaussian distribution.

- For each 3D Gaussian point, we calculate its weight $w$ based on the distance $d$, using the formula: $w = 2 \times \text{sigmoid}(-d \times a)$. This results in a monotonically decreasing relationship between $w$ and $d$; that is, the farther a point is from the landmark, the smaller its weight $w$. Conversely, $1 - w$ increases with $d$.

- As shown in Eq. 8, when the randomly sampled edit strength is larger than or equal to 0.5, indicating a result for stronger edits, the model assigns higher weights $w$ to points closer to the landmarks. Conversely, when the edit strength is smaller than 0.5, lower weights $1 - w$ are assigned to these points.

This landmark-driven loss encourages the model to focus edits on the facial region, mitigating the influence of unwanted gradients caused by small edit strengths that could otherwise result in minimal edits. Note that varying the $\alpha$ value controls the degree of localization in editing (see Fig. 5). We also refer the readers to Fig. 6 for ablation studies on the effectiveness of our proposed landmark-driven loss.

**Motivation behind the weight map function.**   The motivation behind our design of the weight map is three-fold: (1) In prior 3D editing methods, the edit strength is typically sampled from a Gaussian distribution at each iteration. This leads to undesired minimal edits for human head avatars, as gradients from small edit strengths tend to average out those from larger strengths. (2) Simply fixing the edit strength to a constant value, whether large or small, also proves to be ineffective. A large edit strength may cause loss of identity, while a small one often leads to negligible edits. (3) To address this, we thus design a loss function that

emphasizes gradients from stronger edits while still retaining useful information from weaker ones to maintain quality and identity. Our current weight map design is a specific implementation of this idea, which assigns weights based on the distance to 3D landmarks, effectively guiding the model to preserve identity while enabling meaningful edits.

**Explanation for not generating Multi-View Videos from Input Monocular Video**   As mentioned in Sec. 3.2 of the main text, we selected a single frame with a natural facial expression from the monocular video and used V3D to generate multi-view images, rather than converting the entire monocular video into a multi-view video. That is because current techniques Xie et al. (2024) still struggle with head-related tasks, making it impractical to generate multi-view videos from the input monocular video, as shown in Fig. 9.

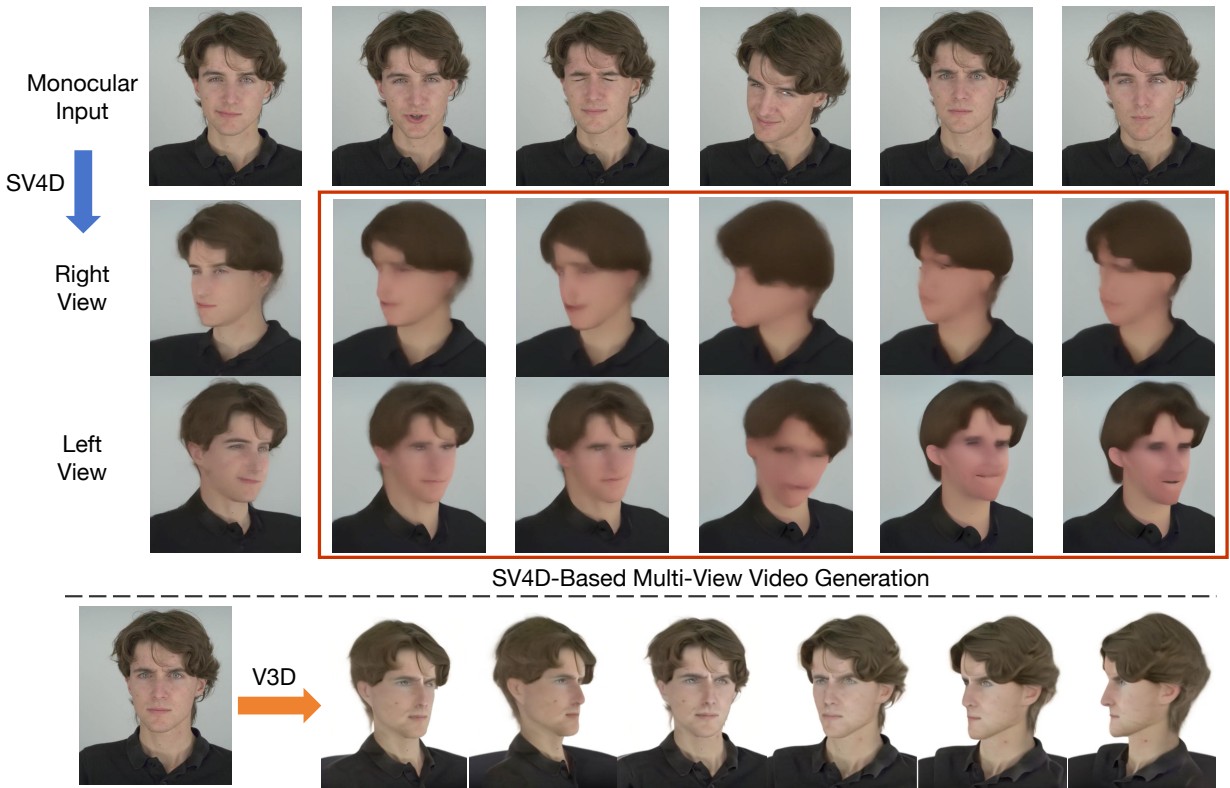

Figure 9: Limitation of Generating Multi-View Videos from Input Monocular Video

**Analysis regarding face model**   We follow our baseline method (Gaussian Head Avatar) to employ BFM for extracting facial parameters, a common practice in recent 3DMM-based approaches. Since we primarily rely on high-level facial parameters, such as landmarks, expression coefficients, and head pose, the finer texture details captured by more recent face models are unlikely to significantly affect our results. To demonstrate our choice, we further perform quantitative comparisons of reconstruction quality across different face models (as in Tab. 6), using 7 examples from the NeRSermble dataset:

# B   Additional Qualitative Comparison

We present additional qualitative results compared to existing methods, as detailed in Sec. 4.2. The results in Fig. 10 demonstrate that our approach not only adheres closely to the given edit prompts but also preserves the person's original characteristics, such as clothing, facial structure, and features.

Table 6: **Quantitative comparisons regarding the employed face model.**

|  | Using FaceVerse (Wang et al., 2022) | Using FLAME (Li et al., 2017a) | Using BFM |
|---|---|---|---|
| LPIPS ↓ | 0.073 | 0.071 | **0.070** |
| SSIM ↓ | 0.880 | **0.884** | **0.884** |
| PSNR ↑ | 27.45 | 27.38 | **27.50** |

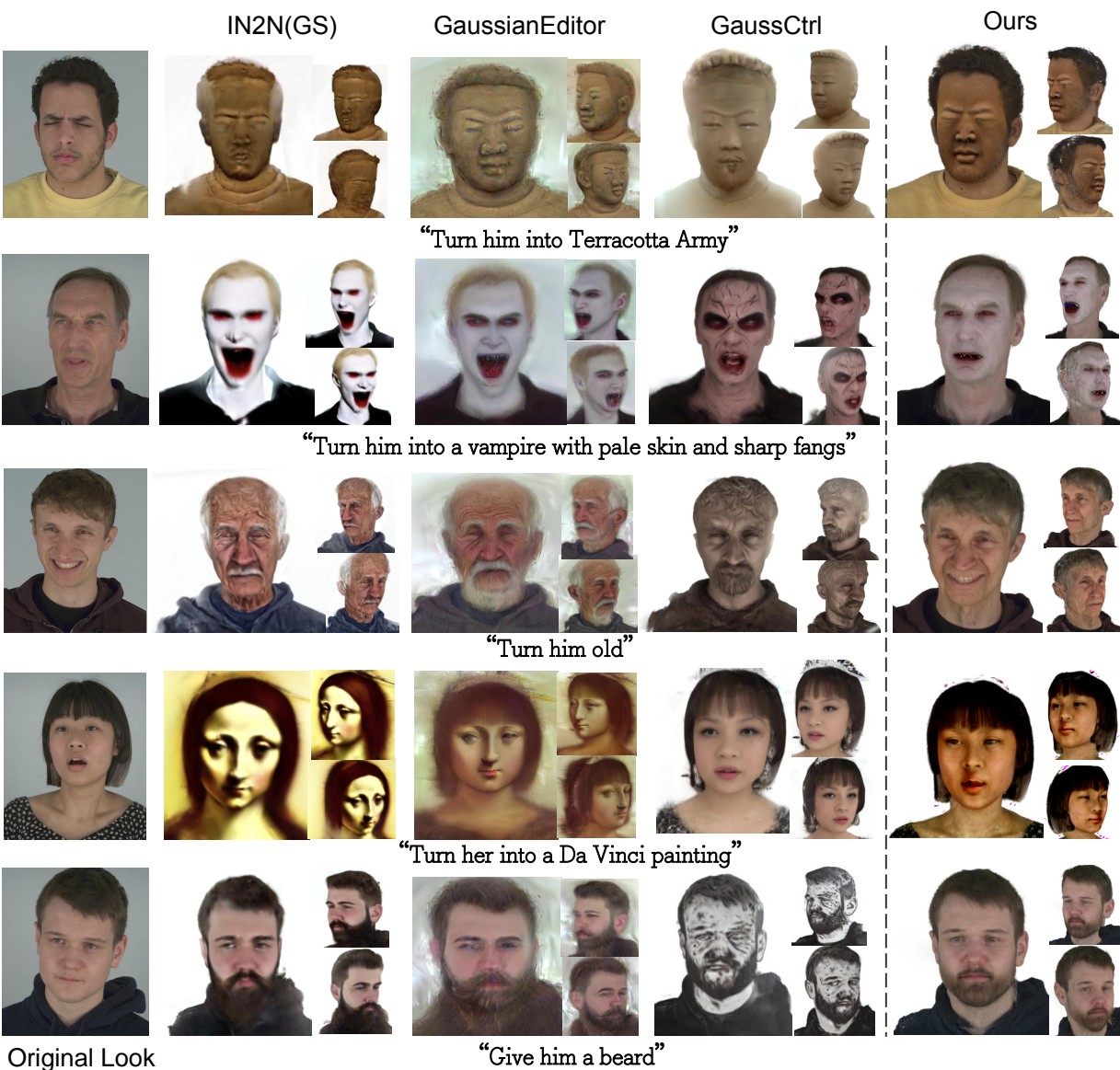

Figure 10: **Additional Qualitative Comparison** with existing 3D scene editing methods.

## C   Additional Animation Results

StyleTalker not only achieves high-quality editing results but can also be driven by arbitrary speakers. Fig 11 showcases the outcomes of both character editing and driving, demonstrating that our results accurately align with the expressions of the target speaker. Furthermore, our outputs exhibit superior clarity and maintain strong multi-view consistency.

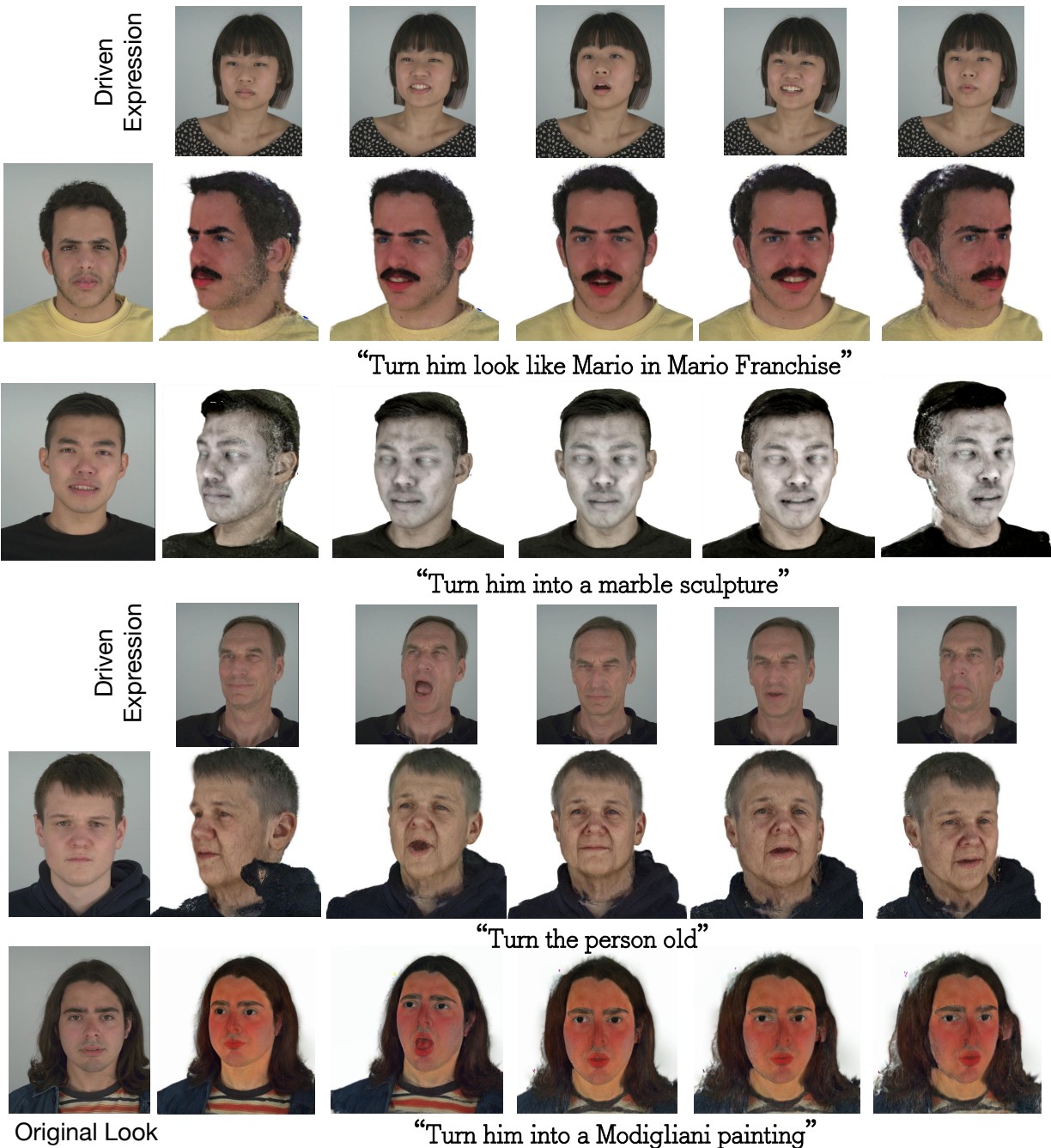

Figure 11: **Talking head editing with different animation results.**

