# OpenReview forum: "StyleTalker: Stylized Talking Head Avatar from Monocular Video"
_TMLR — Rejected by TMLR_

### Review · Reviewer_7EdE · 2025-11-05

**Summary Of Contributions:**

The paper introduces StyleTalker, a novel text-guided framework for editing and animating dynamic 3D talking head avatars from a single monocular video. StyleTalker proposes two core innovations: (i) a mesh-enhanced 3D Gaussian reconstruction that integrates 3D head priors and multi-view video diffusion to enable high-fidelity dynamic reconstruction from monocular input; and (ii) a landmark-driven editing mechanism that modulates editing strength based on proximity to 3D facial landmarks, thereby preserving identity during localized edits. Experiments show that StyleTalker outperforms state-of-the-art methods in both editing quality and animation fidelity.

Strengths:
1. Editing dynamic 3D avatars from a single video is highly relevant for real-world applications like telepresence, AR/VR, and content creation, where multi-view capture is often infeasible.
2. The combination of mesh priors with 3D Gaussian Splatting for reconstruction, and the use of 3D facial landmarks to spatially modulate editing strength, are well-motivated and address clear gaps in current methods.
3. The emphasis on maintaining identity during editing—especially via geometric (landmark-based) guidance rather than relying solely on semantic segmentation or language models—is a significant step forward for avatar editing.
4. Unlike many editing frameworks that produce static results, StyleTalker supports expression-driven animation of edited avatars, enhancing its utility.

Weaknesses:
1. The landmark-driven editing assumes reliable 3D facial landmark estimation from monocular video—a non-trivial task under occlusion, extreme poses, or low resolution. Failure cases in landmark detection could degrade editing performance, but robustness is not discussed.
2. Interactive demos or source code are very important for a deeper understanding of this work.

**Audience:**

Yes

**Audience Explanation:**

This paper provides a comprehensive framework to construct the StyleTalker. Unlike many editing frameworks that produce static results, StyleTalker supports expression-driven animation of edited avatars, enhancing its utility.

**Broader Impact Concerns:**

No.

**Claims And Evidence:**

Yes

**Claims Explanation:**

Core Innovative Designs:
1. Mesh-enhanced 3D Gaussian reconstruction: A two-stage training strategy is proposed to integrate 3D head priors and multi-view video diffusion models into the reconstruction pipeline, improving the accuracy and flexibility of 3D reconstruction from monocular videos.
2. Landmark-driven talking head editing: To address the challenges of localized editing while preserving the original identity, 3D facial landmarks are used as guides to control editing strength across different facial regions. By adjusting the editing strength based on the distance to landmarks, the framework meets text-guided editing requirements while retaining the avatar’s original identity.

**Requested Changes:**

Please see the weaknesses.

---

### Review · Reviewer_i3fX · 2025-11-15

**Summary Of Contributions:**

This paper introduces StyleTalker, a novel framework for text-guided editing and animation of 3D talking head avatars from monocular video. The key innovations include a mesh-enhanced 3D Gaussian reconstruction method that integrates 3D head priors with multi-view diffusion models, and a landmark-driven editing technique that preserves identity by adaptively controlling edit strength based on facial landmarks. Comprehensive experiments demonstrate state-of-the-art performance in editing quality and identity preservation.

**Audience:**

Yes

**Audience Explanation:**

This work makes significant contributions at the intersection of 3D vision, generative models, and digital avatars - areas of substantial interest within the AI and computer vision communities. By addressing critical limitations of prior methods, such as multi-view dependency and identity loss, it enables practical applications in AR/VR, gaming, and virtual telepresence. The novel integration of diffusion models with 3D Gaussian splatting through landmark-guided editing provides valuable insights for advancing controllable 3D content creation.

**Broader Impact Concerns:**

While StyleTalker enables beneficial applications in virtual communication, entertainment, and assistive technologies through accessible avatar creation, it also raises concerns about potential misuse for generating deceptive content (e.g., manipulated videos) without consent, which could contribute to misinformation. It is recommended that future work incorporate safeguards such as watermarking and detection mechanisms, while emphasizing ethical guidelines in the camera-ready version to mitigate these risks.

**Claims And Evidence:**

Yes

**Claims Explanation:**

1.	The paper is well-organized, and the writing is generally clear and concise.
2.	The paper tackles the challenging task of text-guided editing and animation of 3D talking head avatars from monocular video. Its two core contributions: mesh-enhanced 3D reconstruction and landmark-driven local editing, are innovative and practical, directly addressing key limitations of prior work.
3.	The paper provides extensive quantitative evaluations (user studies, CLIP-based metrics, reconstruction quality) and qualitative comparisons, demonstrating superior performance over strong baselines. Ablation studies validate the necessity of each proposed component.

**Requested Changes:**

1.	The paper showcases successful edits but does not thoroughly analyze failure modes or challenging scenarios (e.g., extreme poses, occlusions, diverse ethnicities). A deeper discussion of the method's boundaries and generalization ability would be valuable.
2.	Specify training/inference times and hardware details (e.g., GPU memory) more explicitly to aid reproducibility.
3.	It would be better to briefly address potential misuse of identity-editing technology in the Broader Impact section.

---

### Review · Reviewer_8Eim · 2025-11-27

**Summary Of Contributions:**

### Summary of Contributions

This paper presents three primary contributions.
First, it proposes an editable pipeline capable of reconstructing a 3D facial model from a single monocular video, which is an exciting and meaningful direction. While many prior works rely on multi-view inputs or static scenes, this work aims to achieve high-quality 3D reconstruction and text-based editing from only a single-view video.

Second, the authors introduce a two-stage reconstruction framework that combines a mesh-based deformation field with 3D Gaussian Splatting.
The mesh deformation (initialized via DMTet) provides geometric structure, and a diffusion-based model generates multi-view images that are used to refine color and attribute learning. This combination attempts to leverage the strengths of both mesh-based and neural rendering methods.

Third, the paper proposes a landmark-driven editing mechanism, where edit strength is controlled through landmark distances. This design helps mitigate identity loss and contributes to more controlled and consistent editing results.

### Strengths

- The idea of producing and editing a high-quality 3D avatar from only a single video is practically valuable and represents an interesting problem setting.
- The proposed landmark-based editing approach effectively alleviates identity loss in localized edits.
- The paper includes several experiments and comparisons that help assess the performance of the proposed method.

### Weaknesses

- Although the overall pipeline is intriguing, the novelty appears somewhat limited because it mainly combines existing components from prior works.
- The end-to-end pipeline is computationally expensive, which may hinder real-world applicability; further optimization would be beneficial.
- Text-based editing examples are rather narrow; large semantic changes (e.g., male-to-female transformation) remain unexplored. Such cases raise questions about how well landmark-based control can handle edits requiring structural changes like hair generation.
- Certain methodological details are difficult to follow. Some formulas lack definitions, and the update rules or processing flow could be described more clearly. More explicit definitions and explanations would strengthen the paper.

### Overall Impression

The problem setting is interesting and the topic has strong practical potential. The paper proposes a promising pipeline that could inspire further research. However, despite its appeal, the current version leaves some important aspects underdeveloped, especially regarding clarity, novelty, and practicality. With further refinement and clearer methodological exposition, the work could become significantly more impactful.

**Audience:**

Yes

**Audience Explanation:**

The topic of reconstructing and editing 3D facial avatars from a single monocular video addresses a realistically important and practically motivated problem. With increasing demand for accessible 3D avatar generation in virtual communication, entertainment, and personalized content creation, the general direction explored in this work is certainly relevant to the community.

Moreover, the combination of mesh-based deformation, diffusion-driven multi-view synthesis, and landmark-guided editing touches on challenges that many researchers in computer vision, graphics, and generative modeling actively study. For this reason, the overall problem setting and high-level methodology are likely to attract interest from at least part of the TMLR readership.

That said, it is also worth noting that the current computational cost is extremely high, and the pipeline takes a very long time to produce a final 3D reconstruction. This practical limitation could raise concerns for readers who are interested in deployable or scalable methods.

**Broader Impact Concerns:**

No concern.

**Claims And Evidence:**

No

**Claims Explanation:**

While the paper presents an interesting direction, I do not believe the current submission provides sufficiently accurate or convincing evidence to fully support its claims. Several key components of the proposed pipeline rely on complex combinations of existing methods, yet the empirical validation does not clearly isolate or demonstrate the effectiveness of each part. In particular, the paper lacks ablation studies or controlled comparisons that would clarify the individual contribution of the deformation initialization, the diffusion-based multi-view synthesis, and the landmark-driven editing module.

Additionally, the exposition of the method is often difficult to follow. Many equations are insufficiently defined, and important variables or update rules are missing or ambiguous. As a result, the technical flow of the approach is not clearly articulated, making it challenging to fully understand or reproduce the proposed pipeline based on the paper alone. This lack of clarity further weakens the strength of the empirical claims, as it is not always clear how the described components are implemented or interact with each other.

Finally, while some qualitative examples are appealing, the quantitative results are limited and do not thoroughly substantiate the claimed improvements. The absence of more challenging or diverse editing scenarios also makes it difficult to evaluate the generality of the approach.

**Requested Changes:**

1. Several equations currently omit definitions of key variables or symbols, which makes the methodology difficult to follow. Providing complete definitions for every term would significantly improve clarity.

2. A step-by-step algorithmic summary or pseudocode would greatly help readers understand how the different components interact, and would improve the reproducibility of the method.

3. The current examples focus on relatively small edits, and it would strengthen the work to demonstrate robustness in cases involving substantial semantic or structural changes.

---

### Decision · Action_Editor_9HyL · 2026-01-31

**Recommendation:** Reject

**Additional Comments:**

If the authors wish to submit a revision, they are advised to very carefully address all reviewer criticisms, especially with regards to supporting the claims with accurate, convincing and clear evidence, and also clarifying the methodology (which may also sharpen the claims).

**Audience:**

Yes

**Audience Explanation:**

The topic of reconstructing and editing 3D facial avatars from a monocular video is a realistic and practical problem with popular motivations, and the work's stated contributions are interesting as well, from both a vision perspective and a generative graphics perspective. However, 2/3 reviewers also note that the methodology is not very clear.

**Claims And Evidence:**

No

**Claims Explanation:**

While one reviewer found the technical approach reasonable (though also noted that robustness is not discussed enough), the other two reviewers reported a clear negative assessment: the paper does not clearly show how each component contributes to the overall performance, it does not thoroughly analyze failure modes or challenging scenarios, and the quantitative evaluation does not sufficiently support the claimed improvements. In sum, it appears that the claims made in the submission are not supported by convincing evidence.

**Resubmission Of Major Revision:**

The authors may consider submitting a major revision at a later time.